# Benefit of stereoscopic volume rendering for the identification of pediatric pulmonary vein stenosis from CT angiography

**Michelle Noga**[1,2]*, **Jiali Luan**[3], **Deepa Krishnaswamy**[1,2], **Brendan Morgan**[4], **Ross Cockburn**[1], **Kumaradevan Punithakumar**[1,2]

**1** Department of Radiology & Diagnostic Imaging, University of Alberta, Edmonton, Canada, **2** Servier Virtual Cardiac Centre, Mazankowski Alberta Heart Institute, Edmonton, Canada, **3** Department of Psychiatry, University of Manitoba, Winnipeg, Canada, **4** Department of Anesthesia, Pain Management & Perioperative Medicine, Dalhousie University, Halifax, Canada

* mnoga@ualberta.ca

## Abstract

The use of three-dimensional (3D) technologies in medical practice is increasing; however, its use is largely untested. One 3D technology, stereoscopic volume-rendered 3D display, can improve depth perception. Pulmonary vein stenosis (PVS) is a rare cardiovascular pathology, often diagnosed by computed tomography (CT), where volume rendering may be useful. Depth cues may be lost when volume rendered CT is displayed on regular screens instead of 3D displays. The objective of this study was to determine whether the 3D stereoscopic display of volume-rendered CT improved perception compared to standard monoscopic display, as measured by PVS diagnosis. CT angiograms (CTAs) from 18 pediatric patients aged 3 weeks to 2 years were volume rendered and displayed with and without stereoscopic display. Patients had 0 to 4 pulmonary vein stenoses. Participants viewed the CTAs in 2 groups with half on monoscopic and half on stereoscopic display and the converse a minimum of 2 weeks later, and their diagnoses were recorded. A total of 24 study participants, comprised of experienced staff cardiologists, cardiovascular surgeons and radiologists, and their trainees viewed the CTAs and assessed the presence and location of PVS. Cases were classified as simple (2 or fewer lesions) or complex (3 or more lesions). Overall, there were fewer type 2 errors in diagnosis for stereoscopic display than standard display, an insignificant difference (p = 0.095). There was a significant decrease in type 2 errors for complex multiple lesion cases ($\geq$3) vs simpler cases (p = 0.027) and improvement in localization of pulmonary veins (p = 0.011). Subjectively, 70% of participants stated that stereoscopy was helpful in the identification of PVS. The stereoscopic display did not result in significantly decreased errors in PVS diagnosis but was helpful for more complex cases.

## Author summary

There has been a tremendous increase in three-dimensional (3D) technology in the non-medical and more recently, medical communities. Computed tomography (CT) imaging

**Data Availability Statement:** Data is available can be accessed on Zenodo, https://doi.org/10.5281/zenodo.7093680.

**Funding:** BM and JL were summer students, funded by an Alberta Innovates summer studentship scholarship, and a Women and Children's Health Research Institute summer studentship award respectively. RC was a summer student funded by the University of Alberta Radiology Endowed Fund. The funders had no role in study design, data collection and analysis, decision to publish, or preparation of the manuscript.

**Competing interests:** The authors have declared that no competing interests exist.

is widely used and easy to translate into 3D technology such as printing and virtual reality. Although there are numerous publications about the use of 3D printing and its applications, there is still relatively little information about virtual reality displays and their usefulness in medical practice. Pulmonary vein stenosis (PVS) is an uncommon condition with significant morbidity in children that is frequently diagnosed with CT. Using our homegrown stereoscopic display system for 3D imaging data, we displayed the 3D CT images for PVS. We tested 24 physicians and compared their diagnostic accuracy for PVS for 3D stereoscopic display vs. conventional monoscopic flat screen display. Our study found that overall, there was no significant difference between the two forms of display for diagnostic accuracy when viewing the entire group of cases, however, for complex cases, there was an improvement when using 3D stereoscopic display.

## Introduction

Volume rendering in medical imaging refers to a visual reconstruction technique which renders an image by sampling voxels of a 3D data set, such as computed tomography (CT) or magnetic resonance imaging (MRI) volumetric data [1]. Despite its 3D appearance, the absence of stereopsis, the perception of depth resulting from combining visual stimuli from both eyes, results in the loss of depth cues when volume-rendered images are displayed using 2D screens, the current standard in medical imaging. In contrast, stereoscopic displays allow for stereopsis and convey depth information through binocular vision. Stereoscopy has been shown to enhance the perception of contrast [2]. In the medical context, the stereoscopic display has been shown to improve visual-spatial tasks and visualization [3,4].

There are many newer approaches to the 3D display of congenital heart disease which have been shown to be beneficial, including 3D printing, mixed reality, virtual reality, and augmented reality [5,6,7]. With recent advances in printing technology, 3D printing has been advocated as a useful method of displaying cardiac anatomy for procedural planning [8,9]. While 3D printing has garnered numerous publications, stereoscopic displays are less commonly studied. Stereoscopic display advantages include the requirement for only a 3D display screen or projector, and goggles, and viewing by multiple users at the same time. In contrast, mixed reality, augmented reality and virtual reality displays are mostly limited to one user at a time. Stereoscopic display has been studied in liver CT for surgical planning [10], digital mammography cancer screening, [11], multimodal brain imaging [12] and 3D echocardiography for congenital heart disease diagnosis [13]. Although 3D modalities are widespread, there is relatively little objective information on the potential additive improvement of 3D stereoscopic display to diagnosis in medical imaging, particularly cardiovascular applications.

Pulmonary vein stenosis (PVS) is a significant cardiovascular pathology in which single or multiple pulmonary veins can become narrowed or obstructed. Echocardiography is the first-line modality for evaluation of patients with suspected PVS, however, an additional form of imaging is commonly employed for confirmation prior to intervention, because of limited sonographic visibility. While CT or MRI can be used, CT has higher spatial resolution and excellent diagnostic performance [14,15]. CT volume rendering improved diagnostic accuracy and confidence in pediatric thoracic vascular disorders [16,17,18] and for pediatric proximal pulmonary vein stenosis [19]. Considering the added benefit of volume rendering for this disease, its variability in presentation, and its reliance on CT for diagnosis, PVS is an ideal condition for investigating the added benefit of stereoscopic display of the volume-rendered images.

The objective of this study was to compare the effect of flat screen conventional display with 3D stereoscopic display of volume-rendered CT on physician perception, using pediatric pulmonary vein stenosis as a test diagnosis.

## Methods

This study was a retrospective observational study using retrospectively acquired CT images and physician volunteers as observers. The study was approved by the University of Alberta Human Research Ethics Board.

### CT images and post-processing

CT images of 22 pediatric patients scanned for pulmonary vein stenosis were acquired retrospectively and consecutively from July 2013 to May 2015 from the radiology pictorial archiving and communications system (PACS). Inclusion criteria were all pediatric patients who had at least 1 year of clinical, surgical or interventional cardiology follow up and acceptable CT image quality. Patient studies were excluded if image quality was poor or there was no follow up or if a prior scan had already been enrolled. Patient ages ranged from 3 weeks to 2 years with a mean of 8 months; there were 8 females and 14 males, weighing 2.9 to 12 kg. The patients were scanned on a Siemens Somatom FLASH scanner, scanned at 80 to 100 kV, 20 mAs, CTDI 0.75 to 1.68 mGy.

The axial CT DICOM images were segmented using Aquarius software (TeraRecon, Inc., San Mateo, California). Diagnoses were confirmed by an experienced pediatric cardiac radiologist with access to all CT images (multiplanar reformats, maximal intensity projections, volume rendering) and patient charts (operative findings, follow-up imaging and clinical data). Manual segmentation was used to remove the adjacent structures, primarily the bones, lungs, aorta and larger pulmonary arteries to show the pulmonary veins and left atrium. A standard CT volume rendering transfer function was employed for all data sets for display on our home-grown display system. The volume rendered images were placed on the stereoscopic system for display on flat screen in either 2D or 3D stereoscopic modes (Fig 1). Only the volume rendered datum was presented, and there was no access to standard axial images or multiplanar reformats (Fig 1).

Of the 18 datasets used for the analysis, there were a total of 31 pulmonary vein stenoses or occlusions. There were 4 normal patients, 4 patients with 1 pulmonary vein stenosis or occlusion, 4 patients with 2 pulmonary vein stenoses or occlusions, 5 patients with 3 pulmonary vein stenoses or occlusions and 1 patient with 4 pulmonary vein stenoses or occlusions. Images were stratified into pulmonary vein stenosis (14 patients) and normal (4 patients) and then randomly divided into 2 groups of 9 patients each (9 for each set A/B and set C/D). Sets A and B had the same 9 test cases, and sets C and D had the same 9 test cases (Fig 2). The remaining 4 non-test patient datasets were randomly distributed to one of set A, B, C, or D, thus creating 4 slightly differing groups. The 4 randomly distributed datasets were used for the participant sessions, however, results were not counted, so that the test sets A and B were identical, and test sets C and D were identical as far as the 9 patients in each group used for analysis. The non-test cases were added to decrease the participant's memory of similarity between test sets. Cases were classified as simple (2 or fewer lesions) or complex (3 or more lesions), and distributed evenly within the groups.

### Volume rendering and display

All images were displayed using a Christie Mirage S+6K DLP projector and Starglass screen (58"x78") for both monoscopic and stereoscopic viewing. The direct volume rendering

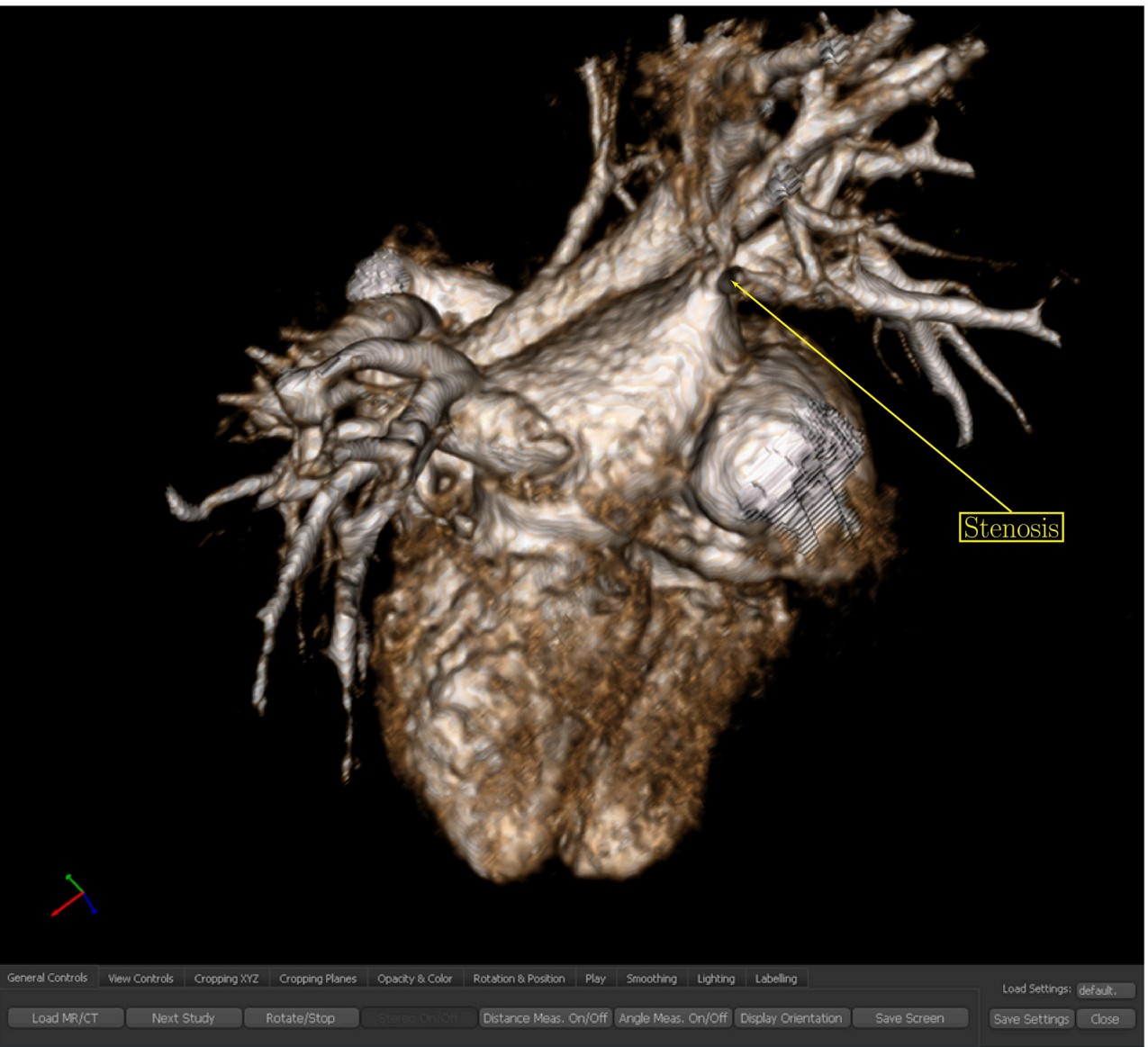

**Fig 1. Volume rendered sample data set of a patient with right pulmonary vein stenosis on the viewing system.** The arrow indicates the area of stenosis.

technique was used for displaying the volumetric CT data [1]. Viewing software was implemented in Python programming language using the Visualization Toolkit (VTK) open-source software module (Kitware Inc., Clifton Park, New York, USA) [20], and Qt graphical user interface library (The Qt Company, Espoo, Finland). The scans were rendered with an NVIDIA Quadro K5200 graphics processor using a ray casting technique, which performs an image-order rendering. All the images were rendered using the same customized color and opacity transfer functions, which were edited by an experienced radiologist, and could not be altered for this study. In the opacity transfer function, higher opacity was assigned for intensity values corresponding to cardiac tissues, and lower opacity was for intensities corresponding to other organs such as lungs to make them transparent. The color transfer function was edited interactively to enhance the aesthetics of cardiac tissues and vessels. The refresh rate for the

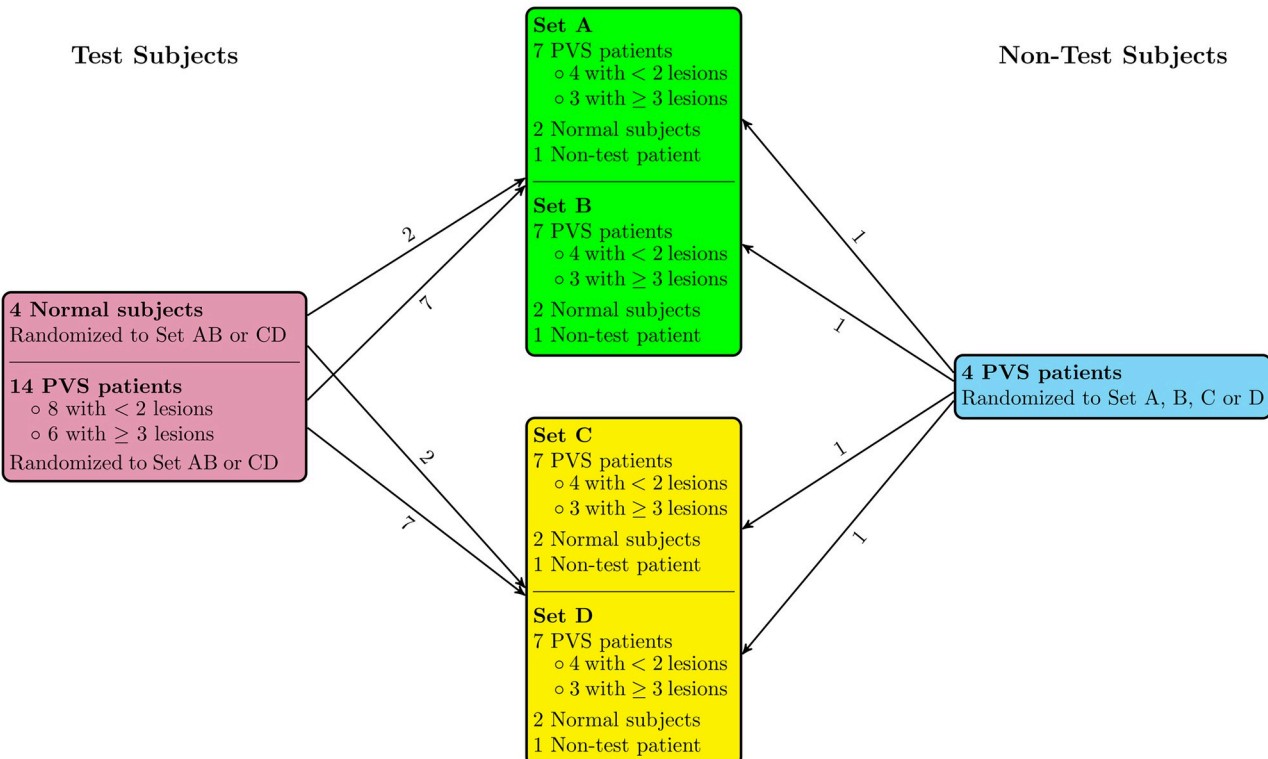

**Fig 2. Distribution of cases into test sets.** Cases of pulmonary vein stenosis (PVS) distributed into 4 test sets, A, B, C or D. Sets A and B have the same test cases. Sets C and D have the same test cases.

project was set to 90 Hz, and no lag in rendering was observed during the interactions such as rotation and scaling. 3D scans were viewed with active shutter glasses, and users manipulated the images using a mouse.

## Participants

Participants were recruited from the departments/ divisions of radiology, pediatric cardiology, pediatric critical care, and cardiovascular surgery. All participants were physicians. Trainees were either residents or fellows. Subject selection assumed appropriate anatomic and pathologic knowledge of pulmonary vein stenosis. All participants signed written informed consent for the study.

## 2D vs 3D image evaluation study

The volunteer study involved a pretest session and 2 sessions for each participant, with the second session occurring at least 2 weeks after the first. In the pretest session, a random-dot stereogram was employed to ensure that participants had stereoscopic perception [21]. A training case was provided to teach basic anatomic positions and to become familiar with using the software to manipulate the images.

For the first testing session, participants were randomly assigned to one of 2 viewing orders: either set A with monoscopic viewing followed by C with stereoscopic viewing or B with stereoscopic viewing followed by D with monoscopic viewing (Fig 3). Participants were not aware of the number of normal, pathological or unscored cases. A participant questionnaire

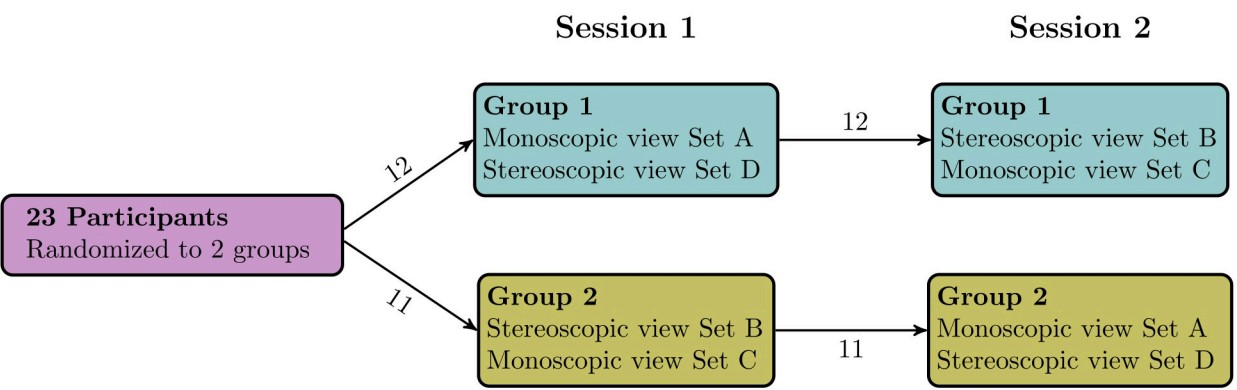

**Fig 3. Viewing order distribution of 4 test sets among the 2 participant groups.**

was administered during the session and included identification of specific pulmonary veins by pointing with a cursor, in addition to identifying pulmonary vein stenosis. Sessions were video-recorded and later reviewed to confirm the accuracy of scoring.

In the second testing session, participants viewed the images in the converse display mode to the first session, viewing the same cases but in the alternate viewing display to the first session (Fig 3). The same questionnaire was administered as in the first session and the session recorded. Participants were not told that 18 of the cases viewed were the same as in the first session but with the converse viewing method.

Errors in identifying pulmonary vein stenosis were reported as type 1 (false positive) or type 2 (false negative) errors. Errors were also recorded if a participant pointed to the wrong vein when asked to localize a specific pulmonary vein. The time that each observer spent on each case was recorded, except for the first case of each set; observers needed significantly more time on the first case of each session to become accustomed to the system.

## Statistics

Statistical analysis was performed using the R programming language [22]. The ggplot2 data visualization package for R was used [23]. In addition, the reshape 2 package for R was also used to aid in data visualization [24]. Mann Whitney two- tailed tests were performed to determine the statistical significance of time spent per case between stereoscopy and monoscopy for each participant because of the absence of 3 time data points. Wilcoxon signed rank two-tailed tests were performed in order to determine statistical significance for type 1, 2 and localization errors between stereoscopy and monoscopy for each participant. P values were also calculated for the other combinations of cases and participants for monoscopic vs stereoscopic views. An alpha level of p value less than 0.05 was set as the threshold for statistical significance.

## Participant questionnaire

The following questions were posed:

- Name the vessel(s) with pulmonary vein stenosis. Point to it.

- Point to the _________ pulmonary vein (specified vessel).

- Do you feel that the addition of 3D visualization aided in your skill performance?

## Results

There were 27 participants, comprised of 15 staff physicians and surgeons, (8 radiologists, 4 cardiologists and 3 cardiac surgeons) and 12 trainees (6 radiology trainees, 5 cardiology trainees, and 1 cardiac surgery trainee). All participants were able to appreciate stereopsis. 3 participants (2 staff physicians and 1 trainee) did not complete the second session; therefore, their results were removed from the study, for a total of 24 participants.

Type 1 errors were calculated retrospectively, and because of data loss, complete data was only available for 19 participants. There were a total of 27 type 1 errors (mean 1.42; median 1; interquartile range (IQR) 2; range 4) for monoscopic display and 32 type 1 errors (mean 1.68; median 2; IQR, 2; range 4) for stereoscopic display (z = -0.584; p = 0.56, Table 1). There were 6 type 1 errors (mean 0.32; median 0; IQR 2; range 2) for monoscopic display and 9 type 1 errors for stereoscopic display (mean 0.47; median 0; IQR 1; range 2) for the complex cases with 3 or more lesions (z = -0.0933, p = 0.35, Table 1). Wilcoxon signed-rank tests indicated no significant difference between groups for type 1 errors.

Overall, there were 292 type 2 errors (false negatives) with monoscopy and 264 errors with stereoscopy (Table 1). There were fewer type 2 errors when stereoscopic images were used for the diagnosis and location of PVS but the result was not significant (z = 1.667, p = 0.10). Participants performed similarly in detecting pulmonary vein stenoses between stereoscopy (mean 11.00; median 10; interquartile range (IQR) 3.25; range 14) and monoscopy (mean 12.17;

**Table 1. Errors in identifying PVS and localization of veins for monoscopic and stereoscopic images.**

| Participant number | Monoscopic type 2 errors (n) | Stereoscopic type 2 errors (n) | Monoscopic type 1 errors (n) | Stereoscopic type 1 errors (n) | Monoscopic localization errors (n) | Stereoscopic localization errors (n) |
|---|---|---|---|---|---|---|
| 1 | 9 | 7 | 4 | 2 | 1 | 0 |
| 2 (trainee) | 8 | 11 | 4 | 4 | 2 | 1 |
| 3 | 12 | 8 | 3 | 2 | 2 | 0 |
| 4 | 13 | 8 | 2 | 2 | 1 | 1 |
| 5 | 10 | 13 | 0 | 0 | 1 | 1 |
| 6 (trainee) | 16 | 15 | 2 | 3 | 3 | 0 |
| 7 (trainee) | 9 | 10 | 0 | 4 | 0 | 0 |
| 8 | 13 | 10 | | | 2 | 0 |
| 9 (trainee) | 11 | 13 | 0 | 0 | 0 | 0 |
| 10 | 9 | 8 | 1 | 3 | 2 | 1 |
| 11 | 10 | 12 | 3 | 4 | 1 | 1 |
| 12 (trainee) | 21 | 21 | 1 | 0 | 5 | 5 |
| 13 | 14 | 9 | 0 | 0 | 0 | 0 |
| 14 | 5 | 8 | 2 | 1 | 0 | 0 |
| 15 (trainee) | 11 | 10 | 0 | 1 | 1 | 0 |
| 16 (trainee) | 8 | 10 | 2 | 1 | 4 | 0 |
| 18 (trainee) | 10 | 10 | 1 | 2 | 1 | 0 |
| 20 (trainee) | 13 | 13 | 2 | 0 | 1 | 0 |
| 21 (trainee) | 16 | 12 | 0 | 2 | 1 | 1 |
| 22 (trainee) | 16 | 12 | 0 | 1 | 1 | 2 |
| 23 | 11 | 11 | | | 0 | 1 |
| 24 | 15 | 9 | | | 9 | 1 |
| 26 | 12 | 11 | | | 0 | 1 |
| 27 | 20 | 13 | | | 1 | 2 |
| Total | 292 | 264 | 27 | 32 | 40 | 18 |

**Table 2. Errors in identifying PVS and localization of veins for monoscopic and stereoscopic images for complex scans with 3 or more PVS lesions.**

| Participant number | Monoscopic type 2 errors (n) | Stereoscopic type 2 errors (n) | Monoscopic type 1 errors (n) | Stereoscopic type 1 errors (n) | Monoscopic localization errors (n) | Stereoscopic localization errors (n) |
|---|---|---|---|---|---|---|
| 1 | 7 | 4 | 1 | 0 | 1 | 0 |
| 2 (trainee) | 5 | 5 | 1 | 1 | 2 | 0 |
| 3 | 8 | 4 | 1 | 1 | 1 | 0 |
| 4 | 8 | 5 | 2 | 1 | 1 | 1 |
| 5 | 6 | 8 | 0 | 0 | 1 | 1 |
| 6 (trainee) | 11 | 9 | 0 | 0 | 3 | 0 |
| 7 (trainee) | 5 | 6 | 0 | 0 | 0 | 0 |
| 8 | 7 | 6 | | | 1 | 0 |
| 9 (trainee) | 8 | 9 | 0 | 0 | 0 | 0 |
| 10 | 5 | 4 | 0 | 1 | 1 | 0 |
| 11 | 6 | 7 | 0 | 1 | 0 | 0 |
| 12 (trainee) | 15 | 12 | 0 | 0 | 1 | 1 |
| 13 | 8 | 6 | 0 | 0 | 0 | 0 |
| 14 | 4 | 7 | 0 | 0 | 0 | 0 |
| 15 (trainee) | 6 | 4 | 0 | 1 | 1 | 0 |
| 16 (trainee) | 4 | 5 | 0 | 0 | 2 | 0 |
| 18 (trainee) | 5 | 4 | 0 | 1 | 1 | 0 |
| 20 (trainee) | 7 | 8 | 1 | 0 | 0 | 0 |
| 21 (trainee) | 11 | 9 | 0 | 1 | 1 | 1 |
| 22 (trainee) | 10 | 6 | 0 | 1 | 1 | 2 |
| 23 | 7 | 7 | | | 0 | 1 |
| 24 | 9 | 5 | | | 3 | 1 |
| 26 | 7 | 8 | | | 0 | 1 |
| 27 | 14 | 6 | | | 1 | 1 |
| Total | 184 | 155 | 6 | 9 | 22 | 10 |

median 11; IQR 4.5; range 16) for all cases. For complex multi-lesion cases, there were 184 type 2 errors for monoscopic display vs 155 errors for stereoscopic display (Table 2). Complex multi-lesion case type 2 errors accounted for 63% of errors for monoscopic display and 58% of errors for stereoscopic display. The accuracy of diagnosis in complex multi-lesion ($\geq$3) scans, participants performed better (z = 2.214, p = 0.03) with stereoscopy (mean 6.46; median 6; IQR 3; range 11) than monoscopy (mean 7.67; median, 7; IQR, 2.5; range, 11) (Table 3).

**Table 3. Summary of type 2 errors by type of case (all cases, complex, simple), participant type (all, trainee, experienced), and p values.**

| Case type | Participant type | Monoscopic errors | Stereoscopic errors | p value |
|---|---|---|---|---|
| All cases | All participants | 292 | 264 | 0.10 |
| Complex cases | All participants | 183 | 154 | 0.03 |
| Simple cases | All participants | 109 | 110 | 0.95 |
| All cases | Trainee | 139 | 137 | 0.94 |
| Complex cases | Trainee | 87 | 77 | 0.12 |
| Simple cases | Trainee | 52 | 60 | 0.16 |
| All cases | Experienced | 153 | 127 | 0.07 |
| Complex cases | Experienced | 96 | 77 | 0.11 |
| Simple cases | Experienced | 57 | 50 | 0.15 |

For trainees, there was overall no significant improvement in accuracy with stereoscopy (139 type 2 errors, mean 12.6 errors/trainee) compared to monoscopy (137 type 2 errors, mean 12.5 errors/trainee), with z = 0.071, p = 0.94 (Table 3). Although there was a trend to improvement for staff physicians with stereoscopy (127 type 2 errors, mean 9.3), compared to monoscopy (153 type 2 errors, mean 11.8), it was not statistically significant (z = 1.809, p = 0.07) (Table 3).

Participants localized pulmonary veins more accurately on stereoscopy with a total of 18 errors (mean 0.75; median 0; IQR 1; range 9) compared with monoscopy with 40 errors observed (mean 1.667; median 1; IQR 1.25; range 9) (Table 1). This difference was significant, z = 2.337, p = 0.01, and localization was also more accurate for complex errors, z = 2.101, p = 0.04, based on the results of Table 2.

Time spent viewing each case for monoscopic display was a mean of 65.4 +/- 38.9 seconds, median 57.3 seconds, range 12.5 to 260.2 seconds (S1 Table). For stereoscopic viewing, the mean time was 66.9 +/- 35.4 seconds, median 60.1 seconds, range 11.3 to 257.5 seconds (S1 Table). The time difference between the two types of displays was insignificant (Mann Whitney $U_1$ = 69170.5, $U_2$ = 77133.5, $n_1$ = 381, $n_2$ = 384, p = 0.10, two-tailed). Time was not recorded for 3 out of the 768 cases observed as participants experienced outside interruptions during their sessions.

A majority of participants (16 of 23, 69.5%), stated that stereoscopy helped them in the identification of PVS, with greater confidence in the diagnosis. One participant recorded no response to this question.

## Discussion

This study represents one of the few objective studies measuring the effect of 3D stereoscopic display on clinician perception in cardiovascular imaging, in this case, for the diagnosis of pulmonary vein stenosis. While there has been a previous study showing an advantage for 3D volume rendering over standard multiplanar reconstruction for this disease [19], no study has assessed the advantage of 3D stereoscopic display for volume-rendered 3D images in assessment of pulmonary vein stenosis. Our study has shown improved diagnostic accuracy and confidence in recognition of pulmonary vein stenosis, and improvement in anatomic localization of pulmonary veins with stereoscopic display.

There was a trend toward improved accuracy of diagnosis for all pulmonary vein stenosis cases, which was not statistically significant, and likely reflected the large number of type 2 errors for complex cases. There was an improvement in accuracy and confidence of diagnosis of pulmonary vein stenosis with fewer type 2 errors for the more complex diagnostic cases with multiple lesions. These results are consistent with other studies that show an improvement in diagnosis using stereoscopic 3D displays [13,25]. Liu showed that the accuracy of detection of cerebral aneurysms was the same (100%) for stereoscopy and conventional CT workstation display for aneurysms over 3 mm; however, it improved with aneurysms less than 3 mm (94% for stereoscopy and 82% for conventional workstation) [25]. Harake et al found that clinicians preferred viewing 3D echocardiography images of both simple and complex lesions in stereoscopic display, as evidenced by more time and interaction spent viewing stereoscopic display when both displays were equally available for viewing [13]. A larger study may have helped to distinguish whether there was an improvement in perception for all types of pulmonary vein stenosis or whether the advantage occurs primarily with complex pathology. Further study would also be required to determine whether the perception advantage for 3D stereoscopic display is dependent on the type of 3D imaging and the specific pathology.

There was an improvement in localization of pulmonary veins with stereoscopic display. This localization task tested the participants' spatial orientation. In diagnosis, correctly locating pathology is as important as recognizing the pathology. If participants were disoriented, they may have recognized pulmonary vein pathology but incorrectly identified the location, resulting in more type 2 errors. The decision to separate the PVS cases into simple and complex was not based on a standard classification. However based on our observation, it became much more difficult for participants to track the position of stenoses spatially as the number of stenoses increased. This assertion is supported by the fact that 63% of type 2 errors with monoscopic display and 59% of type 2 errors in stereoscopic display were attributed to complex cases, while complex cases only comprised 33% of the overall number of cases.

There was a subjective preference for stereoscopy in our study, similar to Harake et al [13]. The preference was not quantitated, and the subjective impression is difficult to measure. This preference is also supported by a few studies that show that 3D printing, which is more widely studied for cardiothoracic surgery, aids in the understanding of spatial relationships, although these studies also cannot quantitate that preference [7,26,27]. Some studies have cited a greater preference on the part of trainees for 3D printing in pre-operative planning, although preference was not quantitated [28].

Results from trainees were compared to staff physicians and surgeons in this study, but there was no significant difference in error rates between these two groups. In general, staff physicians and surgeons showed a larger improvement with stereoscopy than monoscopy compared to the trainees, although this difference was not statistically significant. This would be an area for further study, to determine whether there are differences between experienced and inexperienced viewers for stereoscopy and monoscopy. The role of 3D printing has been more studied, and like stereotactic visualization, some studies have cited a greater preference on the part of trainees for viewing 3D printing in pre-operative planning [28]. For surgical training, virtual reality training was not as optimal as wet lab simulation but has been shown to present a reasonable alternative to wet lab simulation [29].

The ground truth was determined by a single individual which could bias the study results. However, the individual had access to all follow up surgical, catheterization, imaging and clinical chart data to confirm diagnoses. While confirming ground truth by a team of individuals would have been useful, the follow up data was likely more objective as a measure of ground truth.

A limitation of our study was the use of only volume rendered images to make a diagnosis of pulmonary vein stenosis. Usually in clinical practice, physicians view conventional multiplanar reconstructions extensively and therefore may have made more type 2 errors than usual. The use of volume rendered images only may also account for the similar performance of staff physicians to trainees, as staff physicians are accustomed to using typical multiplanar images for this diagnosis. Participants were deliberately not shown the conventional multiplanar reconstructions so that they would not be biased by their impression from the 2D planar reconstructions. By showing only volume rendered images, we attempted to evaluate only the effect of the different display modes on accuracy of diagnosis, not the optimal diagnostic pathway in a real world clinical situation. We are not suggesting that volume rendering replace viewing of multiplanar reconstructions for diagnosis, but merely evaluating the 2 viewing modes objectively. In addition, there were few type 1 errors, which may have been a bias of the exclusive volume rendering technique, as this was the case for monoscopy and stereoscopy. In other words, with volume rendering, observers had no difficulty in recognizing normal vessels.

We split the cases into 2 sessions, so that cases would be shown to a participant as either monoscopy first and then stereoscopy or vice versa, resulting in a form of internal control. A limitation of this method was the need to show the same images and cases to the participants

in the second session, but in the alternate viewing mode, with the possibility of participants remembering cases. The second session was performed at least 2 weeks after the first session to decrease bias from the participants remembering a given case from the first session. The participants were not informed that the cases were the same between sessions. We also introduced one non-test case into each set of cases, so that only 9 out of 10 cases in a test batch were scored in an attempt to decrease the participants' ability to recall previous cases.

Time spent per case was not significantly different for monoscopic and stereoscopic display. It might be expected that time would be shorter for stereoscopy however, a similar study by Harake et al noted that participants spent longer looking at stereoscopic images as they perceived that there was more information available [13].

User interaction in this study was limited to rotation, panning, and zooming. The ability to move and rotate the image has been shown to enhance perception of 3D nonmedical objects regardless of whether monoscopic and stereoscopic displays were used, with use of rotation and monoscopic display being superior to static stereoscopy along [30]. The effect of manipulation on stereoscopy is difficult to isolate in our study, however, subjects were allowed equal freedom to manipulate the object for both display modes in our study. Therefore, monoscopic display acted as a control for stereoscopic display.

An additional limitation of the study was that the results are not generalizable to other forms of congenital heart disease or other pathology. However, the study did compare monoscopic display to 3D stereoscopic display of complex thoracic anatomy in an objective manner. While the results are similar to other studies, additional objective studies are required to determine the role of stereoscopic display of volume rendered imaging on perception in medical imaging. No cost benefit analysis was performed in this study, and this type of analysis should be performed prior to routinely using this type of technology in medical imaging [9].

## Conclusion

We have completed a retrospective blinded observational study to objectively compare the effect of the stereoscopic 3D display to the monoscopic conventional display of volume-rendered CT images in the setting of pediatric pulmonary vein stenosis. There was no significant overall improvement in the identification of lesions when using stereoscopic 3D display, however, 70% of users reported greater subjective confidence, while objectively, users localized pulmonary veins more accurately and displayed improved accuracy for complex multi-lesion cases of pulmonary vein stenosis. Further studies of other diseases would be useful in determining the potential benefit and role of stereoscopic display for medical imaging diagnosis.

## Supporting information

**S1 Table. Time (in seconds) spent by each participant per case.**
(CSV)

## Acknowledgments

Hardware for the study was provided by an unrestricted charitable grant from Servier Canada through the University of Alberta Hospital foundation.

## Author Contributions

**Conceptualization:** Michelle Noga, Kumaradevan Punithakumar.

**Data curation:** Jiali Luan, Brendan Morgan, Ross Cockburn.

**Formal analysis:** Deepa Krishnaswamy, Kumaradevan Punithakumar.

**Investigation:** Michelle Noga, Brendan Morgan, Ross Cockburn, Kumaradevan Punithakumar.

**Methodology:** Michelle Noga, Jiali Luan, Brendan Morgan, Kumaradevan Punithakumar.

**Project administration:** Michelle Noga, Kumaradevan Punithakumar.

**Resources:** Michelle Noga.

**Software:** Kumaradevan Punithakumar.

**Supervision:** Michelle Noga, Kumaradevan Punithakumar.

**Validation:** Deepa Krishnaswamy.

**Writing – original draft:** Michelle Noga, Jiali Luan.

**Writing – review & editing:** Michelle Noga, Jiali Luan, Deepa Krishnaswamy, Brendan Morgan, Ross Cockburn, Kumaradevan Punithakumar.

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
