## [Decision Letter · Decision Letter 0]

31 Dec 2021

PDIG-D-21-00111

Benefit of Stereoscopic Volume Rendering for the Identification of Pediatric Pulmonary Vein Stenosis from CT Angiography

PLOS Digital Health

Dear Dr. Noga,

Thank you for submitting your manuscript to PLOS Digital Health. After careful consideration, we feel that it has merit but does not fully meet PLOS Digital Health's publication criteria as it currently stands. Therefore, we invite you to submit a revised version of the manuscript that addresses the points raised during the review process.

We look forward to receiving your revised manuscript.

Kind regards,

Judy Wawira Gichoya

Section Editor

PLOS Digital Health

Journal Requirements:

1. Please amend your detailed Financial Disclosure statement. This is published with the article, therefore should be completed in full sentences and contain the exact wording you wish to be published.

State what role the funders took in the study. If the funders had no role in your study, please state: “The funders had no role in study design, data collection and analysis, decision to publish, or preparation of the manuscript.”

2. Please ensure that the funders and grant numbers match between the Financial Disclosure field and the Funding Information tab in your submission form. Note that the funders must be provided in the same order in both places as well.

3. Please update the completed 'Competing Interests' statement. If you have no competing interests to declare, please state “The authors have declared that no competing interests exist”.

4. Please provide a complete Data Availability Statement in the submission form, ensuring you include all necessary access information or a reason for why you are unable to make your data freely accessible. Note that it is not acceptable for the authors to be the sole named individuals responsible for ensuring data access.

PLOS defines a study's minimal data set as the underlying data used to reach the conclusions drawn in the manuscript and any additional data required to replicate the reported study findings in their entirety. Any potentially identifying patient information must be fully anonymized. 

If your research concerns only data provided within your submission, please write "All data are in the manuscript and/or supporting information files" as your Data Availability Statement.

5. Please provide separate figure files in .tif or .eps format only and remove any figures embedded in your manuscript file. Please ensure that all files are under our size limit of 20MB. 

For more information about how to convert your figure files please see our guidelines: https://journals.plos.org/digitalhealth/s/figures

Additional Editor Comments (if provided):

Thank you for your work. As noted , the work requires extensive revision to make it relevant to the readership. Please consider the detailed feedback provided by our reviewers to ensure that the work is improved 

Reviewers' comments:

Reviewer's Responses to Questions

**Comments to the Author**

1. Does this manuscript meet PLOS Digital Health’s publication criteria? Is the manuscript technically sound, and do the data support the conclusions? The manuscript must describe methodologically and ethically rigorous research with conclusions that are appropriately drawn based on the data presented.

Reviewer #1: Yes

Reviewer #2: No

2. Has the statistical analysis been performed appropriately and rigorously?

Reviewer #1: No

Reviewer #2: No

3. Have the authors made all data underlying the findings in their manuscript fully available (please refer to the Data Availability Statement at the start of the manuscript PDF file)?

Reviewer #1: Yes

Reviewer #2: No

4. Is the manuscript presented in an intelligible fashion and written in standard English?

Reviewer #1: Yes

Reviewer #2: No

5. Review Comments to the Author

Reviewer #1: PDIG-D-21-00111 stereoscopic volume rendering CT

Overall: This study reports on assessment of performance using volumetric images either in 2D or 3D mode for detection pediatric pulmonary vein stenosis from CT angiography.

Abstract: Fine as written.

1) Line 4: Suggest using untested rather than unproven.

Introduction: Overall fine as written in terms of summarizing the problem area.

1) Could do a better job of reviewing prior literature – there were a number of studies for example by Getty et al. looking at stereoscopic display of breast images.

Methods: A few points require clarification.

1) Line 83: What were the inclusion & exclusion criteria for including cases in the study?

2) Line 89: Verify – the gold standard/truth was only determined by a single person? If so this should be noted as a limitation as could be bias.

3) Line 96: Data were not data was as data is plural and datum singular.

4) Line 98: It is not clear why the images were split into 2 sets rather than having all the observers read all the cases.

5) Line 98: If the cases were randomly divided then there was no way to assure equivalence of cases between the 2 groups of cases. Why not randomize with stratification to at least ensure were equal numbers of normal cases in reach group?

6) Line 103: Should explain here rather than later the point of these extra 4 cases and why not counted in the analyses.

7) Line 107: Provide a reference for or explain in more detail what is meant by “direct volume rendering technique”.

8) Line 113: The readers in the study could not adjust color or opacity? How do you know the single person who set them wasn’t biased in some way? Was this the same person that established truth or another expert?

9) Seems like a very useful variable to have measured would have been time to render the decisions. Some prior studies showed differences in time using 2D vs stereoscopic 3D – time often ends up being the far more important factor when someone decides whether to use a given display or mode vs another. Do you have timing data you could report?

10) Line 150: How did they point to things? With a cursor?

Results: A few points require clarification.

1) Line 164: Need to clarify here & in tables 2 & 3 what types of errors they were for each reader – clearly there are FNs but seems like there are likely FPs as well and this needs to be separated & reported on as could be significant differences in each as a function of 2D vs 3D as well as experience level.

2) Line 167: Why using the median here? Were distributions skewed?

3) Line 168-169: FN or FP errors?

Conclusions: Overall fine in terms of summarizing the results, discussing in a broader context and noting limitations.

1) Line 194: There are actually more studies on stereoscopic displays in radiology as noted earlier.

2) Line 243: This is not a limitation! Paired studies are actually preferred as they do use the same images thus eliminating a source of bias/confounding as well as increasing statistical power.

3) A limitation is however that you used 2 different sets of images for the readers thus actually introducing potential bias/confounds and statistical power so this does need to be noted as a limitation.

4) The fact that you did not collect timing data should be noted either as a limitation or as a future point to study.

References: A better lit review is required.

Tables & Figures: See points above.

Reviewer #2: PDIG-D-21-00111

Title

Benefit of Stereoscopic Volume Rendering for the Identification of Pediatric Pulmonary Vein Stenosis from CT Angiography

Major comments

1. The Introduction of the work is somewhat lengthy and might benefit from some degree of synthesis. More so, while a three-dimensional visualization might be especially beneficial in a first-line imaging technique such as ultrasound, computed tomography (CT) already allows image reconstruction to some degree and might perhaps lead to less pronounced advantages in this context. Additionally, CT implies the use of ionizing radiations, whereas increasing the accuracy of ultrasound with CT becoming less necessary might yield further benefits considering the young age of pulmonary vein stenosis patients

2. Please consider clarifying the nature of the study, as it appears to be mostly on the retrospective side

3. Allowing readers to only visualize three-dimensional rendering, with no access to biplanar or multiplanar reconstructions might induce a significant disadvantage on non-stereoscopic visualization of such images, thus introducing a potential source of bias in results from the present work. Indeed, the advantage observed for stereoscopic volume rendering in complex cases might be due to this intrinsic potential source of bias

4. The authors might consider clarifying exactly how the study sample was collected, and how cases datasets were divided and presented to individual readers, as this might not appear entirely clear from the manuscript. Inclusion and exclusion criteria, along with relevant time intervals ought to be reported thoroughly. A figure might perhaps be beneficial in this context.

5. T-tests work under the assumption of data normality, which might not be the case in this work due to the paucity and peculiarity of the study sample. Thus, the authors might consider utilizing non-parametric statistics

6. Please consider adding a clear definition of what constitutes an error, and which types of error might have been encountered by the study readers. More so, please consider reporting the extent of each reader’s individual experience

6. PLOS authors have the option to publish the peer review history of their article (what does this mean?). If published, this will include your full peer review and any attached files.

**Do you want your identity to be public for this peer review?** For information about this choice, including consent withdrawal, please see our Privacy Policy.

Reviewer #1: No

Reviewer #2: No

---

## [Decision Letter · Decision Letter 1]

26 Jul 2022

PDIG-D-21-00111R1

Benefit of Stereoscopic Volume Rendering for the Identification of Pediatric Pulmonary Vein Stenosis from CT Angiography

PLOS Digital Health

Dear Dr. Noga,

Thank you for submitting your revised manuscript to PLOS Digital Health. After careful consideration, we feel that it does not yet fully meet PLOS Digital Health's publication criteria. Therefore, we invite you to submit a revised version of the manuscript that addresses the points raised during the review process, particularly the points raised by Reviewer #2.

Please submit your revised manuscript within 60 days Sep 24 2022 11:59PM. If you will need more time than this to complete your revisions, please reply to this message or contact the journal office at digitalhealth@plos.org. Please include the following items when submitting your revised manuscript:

We look forward to receiving your revised manuscript.

Kind regards,

Judy Wawira Gichoya

Section Editor

PLOS Digital Health

Journal Requirements:

Additional Editor Comments (if provided):

There are a number of key concerns that the reviewers have raised, and I advise the authors to go through them carefully and address them to strengthen the manuscript.

Reviewers' comments:

Reviewer's Responses to Questions

**Comments to the Author**

1. If the authors have adequately addressed your comments raised in a previous round of review and you feel that this manuscript is now acceptable for publication, you may indicate that here to bypass the “Comments to the Author” section, enter your conflict of interest statement in the “Confidential to Editor” section, and submit your "Accept" recommendation.

Reviewer #1: All comments have been addressed

Reviewer #2: (No Response)

Reviewer #3: (No Response)

2. Does this manuscript meet PLOS Digital Health’s publication criteria? Is the manuscript technically sound, and do the data support the conclusions? The manuscript must describe methodologically and ethically rigorous research with conclusions that are appropriately drawn based on the data presented.

Reviewer #1: Yes

Reviewer #2: No

Reviewer #3: Partly

3. Has the statistical analysis been performed appropriately and rigorously?

Reviewer #1: Yes

Reviewer #2: No

Reviewer #3: Yes

4. Have the authors made all data underlying the findings in their manuscript fully available (please refer to the Data Availability Statement at the start of the manuscript PDF file)?

Reviewer #1: Yes

Reviewer #2: Yes

Reviewer #3: Yes

5. Is the manuscript presented in an intelligible fashion and written in standard English?

Reviewer #1: Yes

Reviewer #2: Yes

Reviewer #3: Yes

6. Review Comments to the Author

Reviewer #1: Authors have adequately addressed my concerns in this extensive revision

Reviewer #2: 1. The Introduction of the work is still somewhat lengthy and might benefit from some degree of synthesis. More so, whereas ultrasound might not be viable, the authors might consider briefly outlining why other imaging techniques such as magnetic resonance imaging (MRI) would not be viable for such purposes

2. While indeed the addition of multiplanar reconstructions would not allow for a stereoscopic vs monoscopic evaluation, this would have been closer to routine clinical practice, and the authors might have considered testing two different image sets with and without stereoscopic data as a further aid. Indeed, in all clinical settings, readers would have access to monoscopic and multiplanar reconstruction datasets

Reviewer #3: This paper seeks to explore the advantages of using stereoscopic displays, rather than a monoscopic view of the same scene, when visualizing volume rendered images of the stenotic pediatric pulmonary vein. Through use studies, the authors conclude that in general there is no statistical difference in the ability of users to identify lesions between conventional 3D display and stereo, the users nevertheless reported greater subjective confidence and demonstrated improved accuracy with complex multi-lesion cases of pulmonary vein stenosis. The authors have made considerable improvements since the first version (upon which this reviewer did not comment).

However, I am confused about the motivation of the study. Is there an un-met need that is being addressed? From my viewpoint the study would carry more weight if it compared multi-planar analysis of the images with both imaging modalities, and in addition to diagnostic accuracy, recorded both the time and cognitive load experienced by the observers. While I understand that Lee et al in 2013 demonstrated the superiority of 3D visualization (not stereo) to multi-planar, it would be helpful to re-enforce their findings with your own. It would also have been interesting to have each user comment on whether the analysis of 3D images was more intuitive and ergonomic that scrolling through the 3D dataset via 2D planes. One feasible scenario would be to employ the 3D visualization mode to rapidly identify suspected locations of stenoses and then to verify them on the basis of multi-planar visualization. 

I note that the quality of the image displayed in the paper seems to be far from state of the art by today’s standards. The images displayed in the cited paper Lee et al. dealing with a similar topic appear to be of much higher quality. State of the art, at least in the Siemens world, involves “cinematic rendering”. 

It would be helpful if the locations of the stenoses in Figs 1 and 2 were identified, and also to comment on the nature of the displayed images – they appear to be very different. Several examples of lesions depicted for different patients would be more helpful than the image of two individuals looking at a screen.

While I agree that stereoscopy seems as if it would be a more natural modality for viewing the 3D images, it is mostly of value when assessing relative depth of objects. I may be missing something here, but I do not believe that depth assessment is actually being employed in the identification of stenoses.

Another aspect that I would like to see discussed more fully is the nature of the manipulations the users accessed during their studies. It is mentioned that the users were able to rotate and scale the images, and if so, does not the rotation also act to provide a similar experience to stereo with respect to identification of depth cues. In fact . in 1991, Sollenberger and Milgram in “A Comparative Study of Rotational and Stereoscopic Computer Graphic Depth Cues” https://doi.org/10.1177/154193129103502007 concluded that rotational displays did a better job than stereo alone (but that the performance was enhanced when using a stereo display with rotation). 

It would seem therefore that the act of manipulation cannot be teased out from the stereoscopic visualization, and may be the reason that there is so little difference in the results. It would perhaps have been more telling if 4 conditions had been analyzed: Static 3D, Rotating 3D, Static Stereo, Rotating Stereo.

The statement that users performed better overall with stereoscopy in the complex cases (>= 3 lesions) (p=.03) is curious, since from my reading of the tables, in neither group alone (p=0.12; 0 .11, for trainees and experts respectively) was this the case. Also given that there were only 6 cases in this category I don’t think any conclusions can regarding differences between mono and stereo can be made. 

Incidentally, in the tables, there is no point in expressing p-values to more than 2 dec places.

I believe Line 190 should read “There were 6 type 1 errors…”

In the supplementary data, the file Type2_and_localization _errors.csv seems to be a duplication of the Time.csv file.

7. PLOS authors have the option to publish the peer review history of their article (what does this mean?). If published, this will include your full peer review and any attached files.

**Do you want your identity to be public for this peer review?** For information about this choice, including consent withdrawal, please see our Privacy Policy. 

Reviewer #1: No

Reviewer #2: No

Reviewer #3: No

---

## [Decision Letter · Decision Letter 2]

10 Feb 2023

Benefit of Stereoscopic Volume Rendering for the Identification of Pediatric Pulmonary Vein Stenosis from CT Angiography

PDIG-D-21-00111R2

Dear Dr. Noga,

We are pleased to inform you that your manuscript 'Benefit of Stereoscopic Volume Rendering for the Identification of Pediatric Pulmonary Vein Stenosis from CT Angiography' has been provisionally accepted for publication in PLOS Digital Health.

Best regards,

Ismini Lourentzou

Section Editor

PLOS Digital Health

Reviewer Comments (if any, and for reference):

Reviewer's Responses to Questions

**Comments to the Author**

1. If the authors have adequately addressed your comments raised in a previous round of review and you feel that this manuscript is now acceptable for publication, you may indicate that here to bypass the “Comments to the Author” section, enter your conflict of interest statement in the “Confidential to Editor” section, and submit your "Accept" recommendation.

Reviewer #4: All comments have been addressed

2. Does this manuscript meet PLOS Digital Health’s publication criteria? Is the manuscript technically sound, and do the data support the conclusions? The manuscript must describe methodologically and ethically rigorous research with conclusions that are appropriately drawn based on the data presented.

Reviewer #4: Yes

3. Has the statistical analysis been performed appropriately and rigorously?

Reviewer #4: Yes

4. Have the authors made all data underlying the findings in their manuscript fully available (please refer to the Data Availability Statement at the start of the manuscript PDF file)?

Reviewer #4: Yes

5. Is the manuscript presented in an intelligible fashion and written in standard English?

Reviewer #4: Yes

6. Review Comments to the Author

Reviewer #4: Reviewers concerns have been met.

7. PLOS authors have the option to publish the peer review history of their article (what does this mean?). If published, this will include your full peer review and any attached files.

**Do you want your identity to be public for this peer review?** For information about this choice, including consent withdrawal, please see our Privacy Policy.

Reviewer #4: No
